# Chinese herbal medicine for children with idiopathic short stature (ISS): A systematic review and meta-analysis

Yingying Li[1], Xinying Chen[2,3,4], Zhengduo Liu[1], Jinghua Yang[2,3,4]*

**1** 2nd Clinical Medical College, Guangzhou University of Chinese Medicine, Guangzhou, Guangdong, China, **2** Department of Pediatrics, Guangdong Provincial Hospital of Chinese Medicine, Guangzhou, Guangdong, China, **3** Lingnan Pediatrics Wen Ziyuan School Studio of Traditional Chinese Medicine, Guangzhou, Guangdong, China, **4** Luo Xiaorong National Famous Traditional Chinese Medicine Expert Inheritance Studio Guangdong, Guangzhou, Guangdong, China

* doumiaomama@126.com

**Data Availability Statement:** All relevant data are within the paper.

**Funding:** The author(s) received no specific funding for this work.

## Abstract

### Background

Idiopathic short stature (ISS) describes a heterogeneous group of children of many unidentified causes of short stature presently without definitive therapy. Chinese herbal medicine (CHM) is an alternative and complementary treatment for children with ISS and has been widely used for ISS while the evidence of its effectiveness is controversial. We conducted this systematic review and meta-analysis in order to evaluate the efficacy of CHM for ISS.

### Methods

PubMed, Embase, Web of science, Sino-Med, Cochrane, CNKI, VIP, and Wangfang Data were electronically searched to collect randomized controlled trials (RCTs) of CHM treatment of ISS from inception to May 2021. Two researchers independently scanned the literature and extracted information on general characteristics, including patient, study design, interventions, and side effects, assessing the CHM intervention's efficacy and the risk of bias. Height, bone age, growth velocity, and IGF-1 level are the main consequences. Height standard deviations score (HtSDS), change in HtSDS (ΔHtSDS), osteocalcin, the peak level of growth hormones (GHP), and predicted adult height (PAH) are the secondary outcomes. Meta-analysis was then performed by using RevMan 5.3 (Cochrane Collaboration).

### Results

Seven articles (569 participants) were included. The Meta-analysis indicated that herbal medicine was associated with increased height (MD 2.16 points; 95%CI, 0.22 to 4.10; P = 0.03), growth velocity (MD 1.47 points; 95%CI, 0.28 to 2.67; P = 0.02), IGF-1 level (MD 28.13 points; 95%CI, 22.80 to 33.46; P<0.00001) and GHP (MD 3.29 points; 95%CI, 1.54 to 5.04; P = 0.0002).

**Competing interests:** The authors have declared that no competing interests exist.

## Conclusion

According to current research, CHM appears to be useful for children with ISS. Due to the limited quality and number of studies included, more high-quality studies are needed to corroborate the above conclusions.

## 1. Introduction

Idiopathic short stature (ISS) described a situation in which in the absence of any systemic, endocrine, nutritional, or chromosomal related disease, the height of a child is less than two standard deviations below the respective mean height for a given age, gender, and population, in a child of normal size and body proportions at birth [1–3]. This definition of ISS mainly includes 70–80% of children with short stature, and there are about 2% of children with ISS in the world [4, 5]. Short stature substantially impacts an individual's employment, marriage, and quality of life. In addition, previous studies showed that short stature might lead to psychosocial problems [6].

The pathophysiology of ISS remains uncertain. Therefore, its standard gold treatment remains open. Several therapies such as pharmacotherapy and physical therapy are available to increase adult height to ease the psychological burden ascribed to short stature in childhood and adult life. The primary approach is growth hormones (GHs) in all the treatments. The US Food and Drug Administration approved GHs for ISS treatment in 2003 [7]. However, because of its varied efficacy, high costs, and negative side effects, the treatment remains controversial [8–11]. Because of these limitations in the usage of GHs, a guideline did not recommend GHs as the therapy of ISS and it is urgent to find other effective treatments [12].

Chinese herbal medicine, as one of the most frequently applied pharmaceutical therapy for ISS, is quite effective in increasing height and growth rate. Pharmacology research proved CHM is effective in increasing height and IGF-1 [13]. Although lots of experimental and clinical research suggested that CHM is beneficial for children with ISS, there are significant differences in the quality between different trials, including some studies with small sample sizes. The intention of the study was to conduct a systematic review and meta-analysis of the literature on the use of CHM for ISS to establish the effectiveness of CHM therapy.

## 2. Methods

This meta-analysis was conducted with the usage of RevMan 5.3 (Cochrane Collaboration) according to the Cochrane Handbook for Systematic Reviews of Interventions [14] and the Preferred Reporting Items for Systematic Reviews and Meta-analyses guidelines were used for the creation of the statement [15]. The protocol for this study was registered on the INSPLAY (Identifier: INPLASY202210034).

### 2.1. Date sources and search strategy

Evidence was gathered by searching eight electronic databases consist of PubMed, Embase, Web of science, Sino-Med, Cochrane, VIP, Wangfang Data, CNKI. We searched databases from their origin until May 2021 and Chinese and English were the only languages available. MeSH terms were used such as "Chinese herbal, Chinese herbal medicine, traditional Chinese medicine, Chinese traditional medicine," and "short stature, idiopathic short stature, dwarfism, growth disorders" to serve as the basis for our search strategy. The reference lists of the including studies were checked to search for additional research.

## 2.2 Study selection criteria

Trials were selected based on the following inclusion criteria: (1) randomized controlled trials (RCTs) whether blinded or not; (2) children with ISS who have clear diagnostic criteria and basis; (3) treatment duration was at least 6 months; (4) a comparison should be done between CHM and other treatment (e.g. CHM versus no treatment, CHM versus other treatment, CHM plus other treatment versus other treatment); (5) at least meeting one of our prespecified outcomes of interest: height, bone age, growth velocity, IGF-1 level, osteocalcin, height standard deviations score (HtSDS), predicting adult height (PAH), change in HtSDS (changes in HtSDS before and after treatment, ΔHtSDS), and growth hormone peak (GHP). (6) language was limited to Chinese and English. Exclusion criteria were (1) trials of participants with other systemic disorders; (2) trials using other traditional Chinese medicine therapies such as massage or acupuncture in the intervention and control groups.

## 2.3 Data extraction

Two investigators (CXY, LYY) retrieved, selected the literature, and did the data extraction from the literature independently. Conflicting information was discussed by the two researchers to reach an agreement or consult a senior investigator. General characteristics, such as patient, research design, interventions, study outcomes, and side effects, are mostly included in the collected information.

## 2.4 Assessment of risk of bias

The risk of bias was evaluated using the evaluation method recommended by Cochrane System Reviewer Manual 5.1 [16], including sequence generation, allocation concealment, blinding, missing outcome data, selective reporting, and other bias. Each domain was categorized into one of three categories: "high risk," "unclear," or "low risk."

## 2.5. Data synthesis and statistical analysis

For continuous outcomes, the data were pooled as mean difference (MD) or standardized mean difference (SMD) with 95% confidence intervals (95%CI) and for dichotomous outcomes, as a risk ratio (RR) with 95%CI. The I-squared ($I^2$) statistic was used to examine heterogeneity between studies in terms of effect measurements. An $I^2 \geq 50\%$ indicated substantial heterogeneity, and we used a random-effects model for meta-analysis. A fixed-effects model was used when the heterogeneity was non-significant ($I^2 < 50\%$). We attempted to evaluate whether the subgroups differed significantly from one another using subgroup analysis, which included treatment duration and specific interventions. RevMan 5.3 (Cochrane Collaboration) was used to conduct the statistical analysis.

# 3. Results

## 3.1 Research selection

The research selection procedure is depicted in Fig 1. A total of 213 titles were identified as potentially relevant to the research project. There remained 29 records after screening. Among them, 22 trials were excluded due to the following reasons: (1) not randomized; (2) no outcomes of interest; (3) other irrelevant interventions or inappropriate control types; (4) no full texts; (5) without a defined diagnosis. Finally, 7 studies satisfied the eligibility requirements.

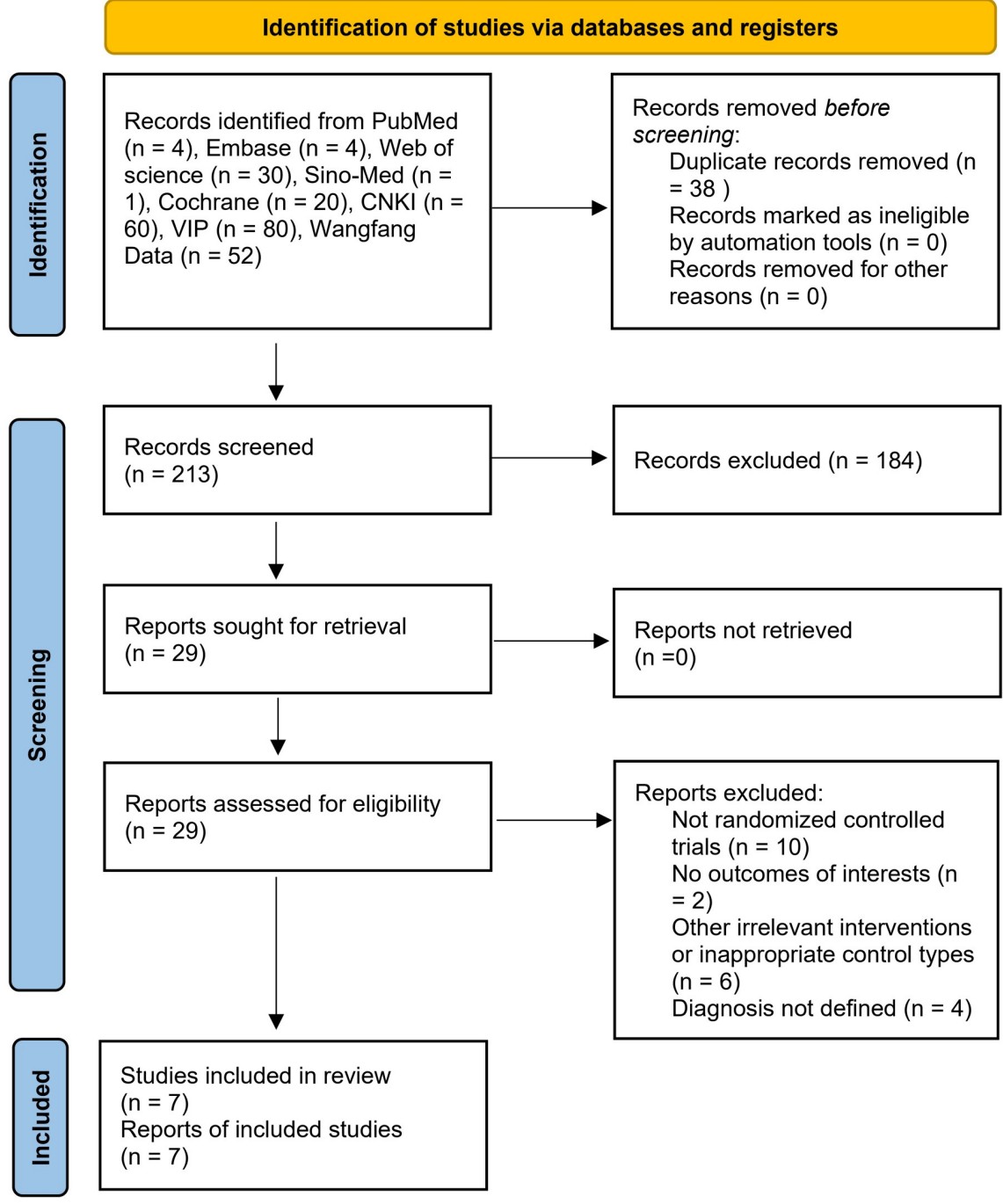

**Fig 1. PRISMA flow chart of study selection in this review.** CNKI: China National Knowledge Infrastructure. VIP: China Science and Technology Journal Database.

## 3.2 Research characteristics

Seven RCTs included 568 participants with ISS, of which 297 children with CHM were used and 271 controls. The age of participants in different treatment groups was similar while the gender of participants ranged because one trial included female participants. Regarding the

treatment duration, three studies [17–19] lasted 6 months while four studies [20–23] followed for 12 months. All studies published between 2010 and 2020 and adopted the diagnostic criteria for ISS formulated by the Chinese Medical Association in 2008 (ISS-2008). The study design of the 7 RCTs is open-label and active-controlled. All CHM formulations were ingested orally, either as a decoction, granule powder, or tablets. Three research [20–22] reported children with ISS treated with CHM in combination with lifestyle interventions, as well as lifestyle interventions alone, two [17, 19] compared herbal medicine plus rhGH with rhGH, and two [18, 23] compared herbal medicine plus nutrient supplement with nutrient supplement alone. Among the studies, Xu 2015 [21] contained two different control groups: children with ISS were treated with rhGH in the first group, while children with ISS were treated with lifestyle interventions in the second group, and Zhou 2012 [23] included two different intervention arms: one treated with nutrient supplement plus herbal medicine and the other treated with herbal medicine alone. In this analysis, all of these studies were adopted. Characteristics of the studies are summarized in Table 1.

## 3.3 The risk of bias

All of the studies stated that participants were assigned to both the herbal medicine intervention and the control groups using "randomization." However, the mechanism of random sequence generation was not specified in 42.8% of studies(n = 3) [21–23]. Four studies utilized the random number table method, all of which had a low risk of bias [17–20]. All of the articles were rated as an uncertain risk because they lacked information about allocation concealment, blinding, selective reporting, and other forms of bias. Because there was no missing data, the majority (85.7%) were considered low-risk. Only one study [18] had six missing outcome data, which put it at high risk. The methodological quality evaluation results of included studies are summarized in Fig 2.

## 3.4 Outcomes of meta-analysis

**3.4.1 Differences in height between groups.** Five studies [17–20, 23] of 438 participants evaluated height outcomes. The random-effects model meta-analysis result indicated that the post-treatment height was significantly higher in the CHM groups than in the control groups. (MD 2.16 points; 95%CI, 0.22 to 4.10; $P = 0.03$; $I^2 = 95\%$) (Fig 3). Subgroup analysis of different treatment durations suggested that the CHM group and CHM in combination with other interventions with a treatment duration of 12 months have a better post-treatment height than that in the control group ($P<0.0001$) (Fig 4).

**3.4.2 Differences in bone age between groups.** Four studies [15–17, 21] of 346 participants assessed bone age. Random effects model meta-analysis results indicated no differences between the herbal treatment groups and the control groups regarding changes in bone age (MD 0.35 points; 95%CI, -0.04 to 0.74; $P = 0.14$; $I^2 = 72\%$) (Fig 5).

**3.4.3 Differences in growth velocity between groups.** In all studies [17–23] of 568 participants, growth velocity was available. Random effects model meta-analysis results indicated that the growth velocity was better in the CHM group than in the control group (MD 1.47 points; 95%CI, 0.28 to 2.67; $P = 0.02$; $I^2 = 100\%$) (Fig 6). Subgroup analysis of different treatment duration showed that with a treatment duration of 12 months, CHM and CHM combined with other treatments increased the growth velocity ($P<0.00001$) (Fig 7). Subgroup analysis of different comparisons indicated that CHM combined with rhGH ($P<0.00001$) and lifestyle intervention ($P<0.00001$) had a better growth velocity than the control groups, but CHM alone (MD -1.65 points; 95%CI, -2.59 to -0.71; $P = 0.0006$) and CHM combined with

**Table 1. Study characteristics in the systematic review and meta-analysis.**

| Author (year) | Country | Study design | Diagnostic criteria | N (C/I) | Age(yr) | Gender (M/F) | Control | Chinese Herbal Medicine | TD (months) | Outcomes | Side effects |
|---|---|---|---|---|---|---|---|---|---|---|---|
| Wang (2020) [17] | China | Open-label, active-controlled | ISS-2008 | 90(45/45) | C: 9.1±1.3 | C: 15/30 | rhGH(0.15–0.20U/kg, qn, ih) | Liujunzi decoction, non-decocting granules (dose adjusted by age and constitution, bid, po) | 12 | a.b.c | No report |
| | | | | | I: 8.7±1.5 | I: 17/28 | | | | | |
| Pan (2020) [18] | China | Open-label, active-controlled | ISS-2008 | 58(28/30) | C: 6.92 ±1.26 | C: 18/10 | Lysine inositol vitamin B12 oral solution(5-10ml, bid, po); Lifestyle intervention | Liujunzi decoction and stomach powder with added flavor (half dose, bid, po) | 12 | a.b.c.d | No report |
| | | | | withdrawn subjects: 6 (2/4) | I: 6.93±0.98 | I: 21/9 | | | | | |
| Feng (2020) [19] | China | Open-label, active-controlled | ISS-2008 | 92(46/46) | C: 12.02 ±0.48 | C: 28/18 | rhGH(0.15U/kg, qn, ih) | Jingui Shenqi Pill (18g, tid, po) | 12 | a.b.c.d.g | Transient headache, red and swollen skin, diarrhea |
| | | | | | I: 11.98 ±0.52 | I:25/21 | | | | | |
| Sun (2017) [20] | China | Open-label, active-controlled | ISS-2008 | 40(20/20) | C: 5.09 ±1.54 | C: 9/11 | Lifestyle intervention | Tonifying and promoting granules (half dose, bid, po) | 12 | a.c.d.e.h.i | No report |
| | | | | | I: 5.37±1.22 | I: 13/7 | | | | | |
| Feng (2014) [22] | China | Open-label, active-controlled | ISS-2008 | 40(20/20) | No mean or standard deviation record (3-13y) | No record of each group | Lifestyle intervention | Different herbal medicine according to the pattern (half dose, bid, po) | 6 | c.d.e.f.i | No report |
| Xu (2015) [21] | China | Open-label, active control | ISS-2008 | 90(30/30/30) | C1:6.78 ±1.44 | C1: 16/14 | Aerobic exercise every day | Ginseng Turtle Feed Particles (half dose, bid, po) | 6 | b.c.e.f.g.i | No report |
| Xu (2015) * [21] | | | | | C2:8.03 ±1.71 | C2: 17/13 | Aerobic exercise every day; rhGH (0.15U/kg, qn, ih); | Ginseng Turtle Feed Particles (half dose, bid, po) | | | |
| | | | | | I:7.98±1.89 | I: 18/12 | | | | | |
| Zhou (2012) [23] | China | Open-label, active-controlled | ISS-2008 | 158(52/53/53) | C:8.52±0.36 | C:0/52 | Lysine(15ml, qn, po) | No treatment | 6 | a.c.d.f | No report |
| Zhou (2012)* [23] | | | | | I1:8.45 ±0.34 | I1:0/53 | No treatment | Zhibaidihuang (8 pills, bid, po) and Dabuyin pill (6g, bid, po) | | | |
| | | | | | I2:8.56 ±0.12 | I2:0/53 | Lysine(15ml, qn, po) | Zhibaidihuang (8 pills, bid, po) and Dabuyin pill (6g, bid, po) | | | |

ISS-2008 = ISS formulated by the Chinese Medical Association in 2008; N = Number of participants; yr = year; TD = treatment duration.

Outcome measures included: a. Height; b. Bone age; c. Growth velocity; d. IGF-1 level; e. HtSDS; f. ΔHtSDS; g. OC; h. GHP; i. PAH.

nutrient supplement ($P = 0.20$) did not increase growth velocity compared with the control groups (Fig 8).

**3.4.4 Differences in IGF-1 between groups.** Five studies [18–20, 22, 23] of 341 participants assessed IGF-1 level. The results of a random-effects model meta-analysis revealed that, when compared to the control groups, herbal medicine could better increase the IGF-1 level (MD 31.78 points; 95%CI, 17.39 to 46.17; $P<0.0001$; $I^2 = 64\%$) (Fig 9). Subgroup analysis of different treatment duration showed that with treatment duration of 12 months ($P<0.00001$)

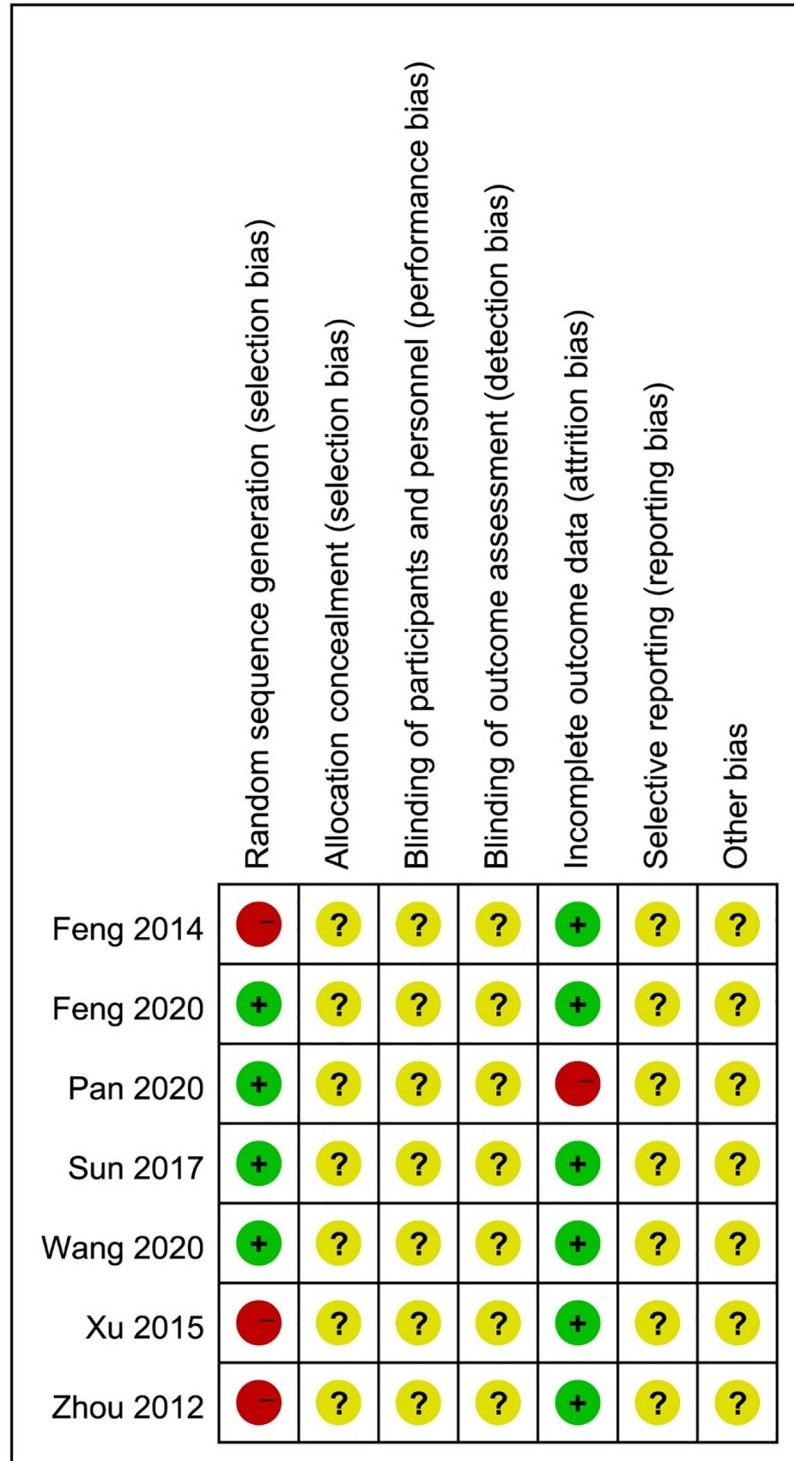

**Fig 2. Risk of bias in included trials.**

and 6 months ($P = 0.04$), CHM and CHM combined with other treatments increased the IGF-1 level (Fig 10). Subgroup analysis of different comparisons indicated that IGF-1 level in CHM plus other drugs($P = 0.001$) and lifestyle intervention ($P = 0.006$) was higher than those in

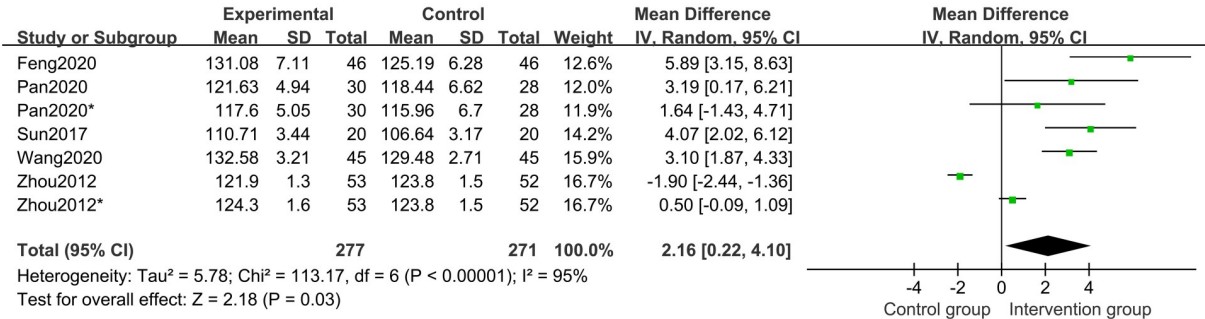

**Fig 3. Forest plot showing the effects of CHM in increasing height.**

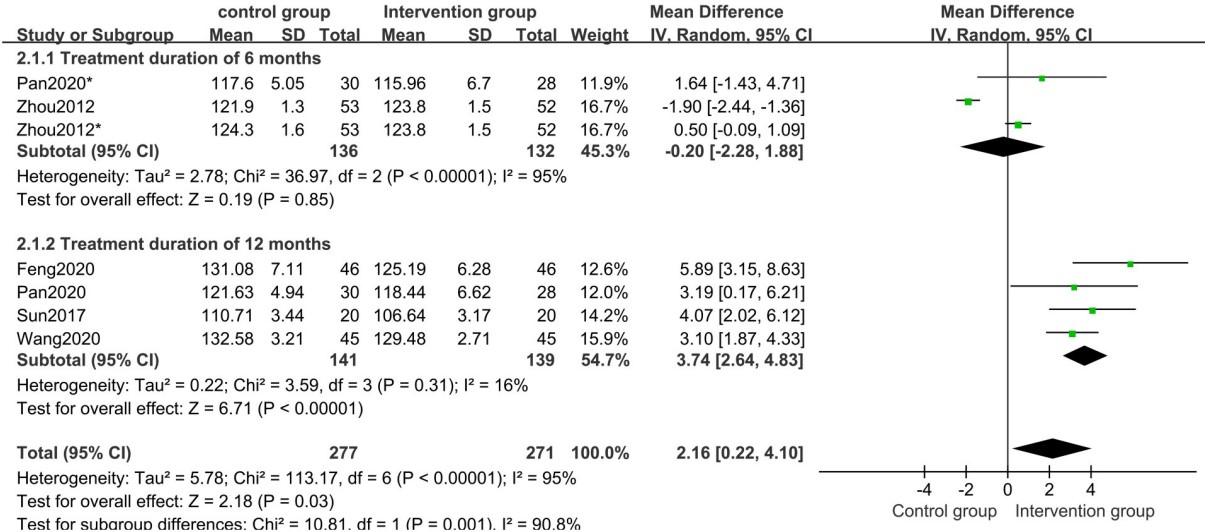

**Fig 4. Subgroup analysis of height based on the treatment duration.**

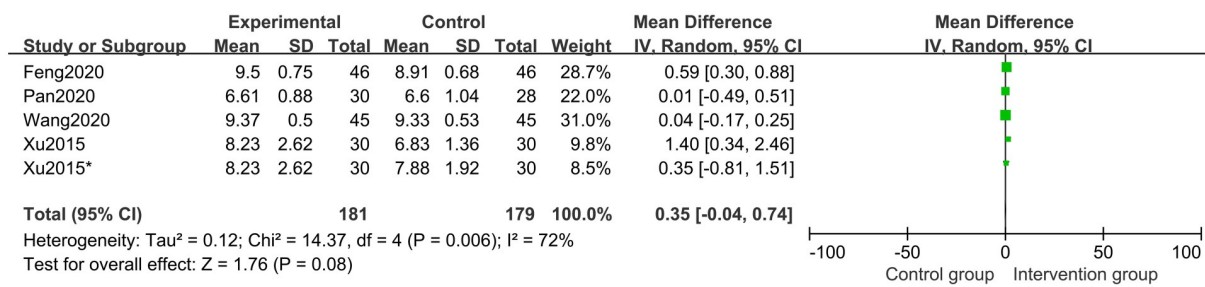

**Fig 5. Forest plot of bone age.**

rhGH and nutrient supplement and lifestyle intervention alone group. Still, the CHM group and the nutrient supplement alone group had no differences ($P = 0.44$). (Fig 11).

**3.4.5 Differences in HtSDS and ΔHtSDS between groups.** Three studies [20–22] of 170 participants described differences in HtSDS. Random-effects model meta-analysis results

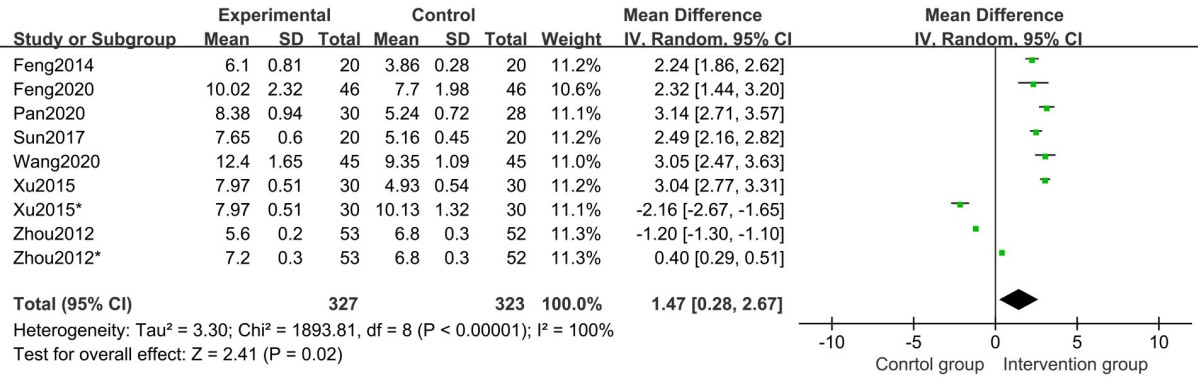

**Fig 6. Forest plot of growth velocity.**

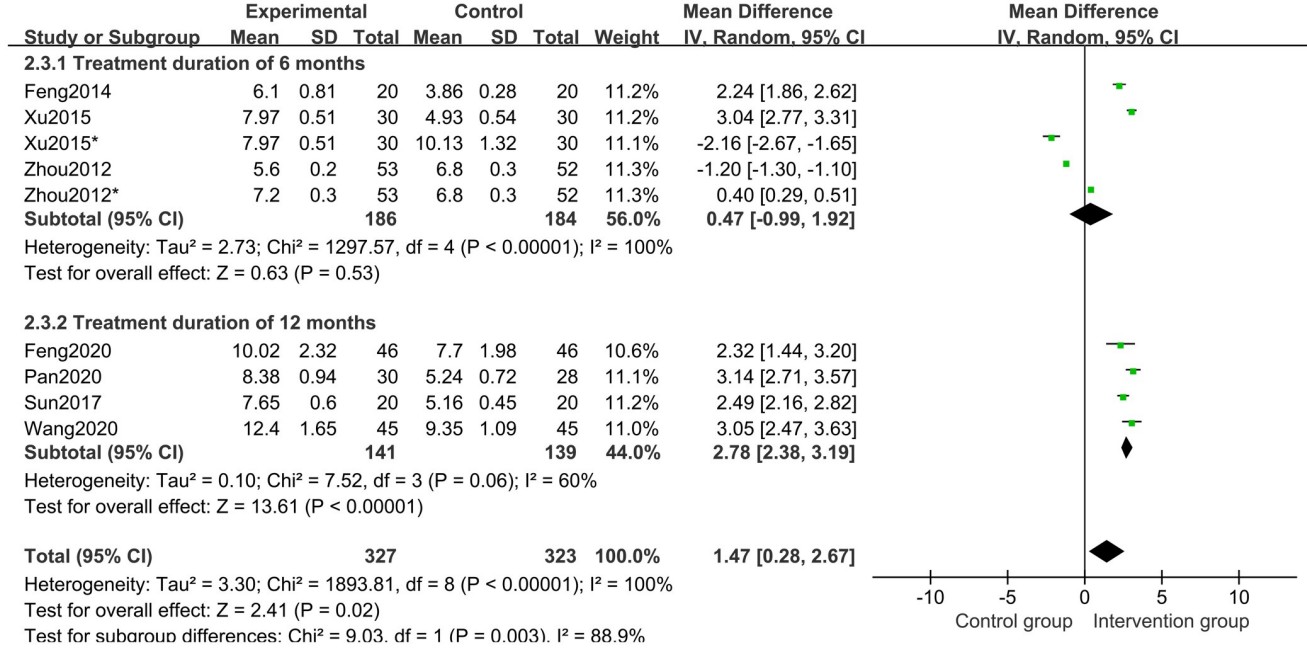

**Fig 7. Subgroup analysis of growth velocity based on the treatment duration.**

suggested that between the HtSDS of the intervention groups and those of the control groups, there were no differences (MD 0.15 points; 95%CI, -0.38 to 0.68; $P$ = 0.58; $I^2$ = 98%) (Fig 12).

Three studies [21–23] of 235 participants reported ΔHtSDS as an outcome. Random-effects model meta-analysis results showed that there were no significant differences in the ΔHtSDS between groups (MD -0.07 points; 95%CI, -0.39 to 0.24; $P$ = 0.65; $I^2$ = 98%) (Fig 13). Subgroup analysis of different comparisons showed that ΔHtSDS in the CHM and CHM plus lifestyle intervention groups is higher than in the control groups ($P$ = 0.001) (Fig 14).

**3.4.6 Differences in osteocalcin and GHP between groups.** Two studies [19, 21] of 182 participants reported osteocalcin as an outcome. The results of a meta-analysis using a random-effects model revealed no significant changes in osteocalcin levels among groups (MD 7.23 points; 95%CI, -0.06 to 14.52; $P$ = 0.05; $I^2$ = 73%) (Fig 15). Subgroup analysis of different

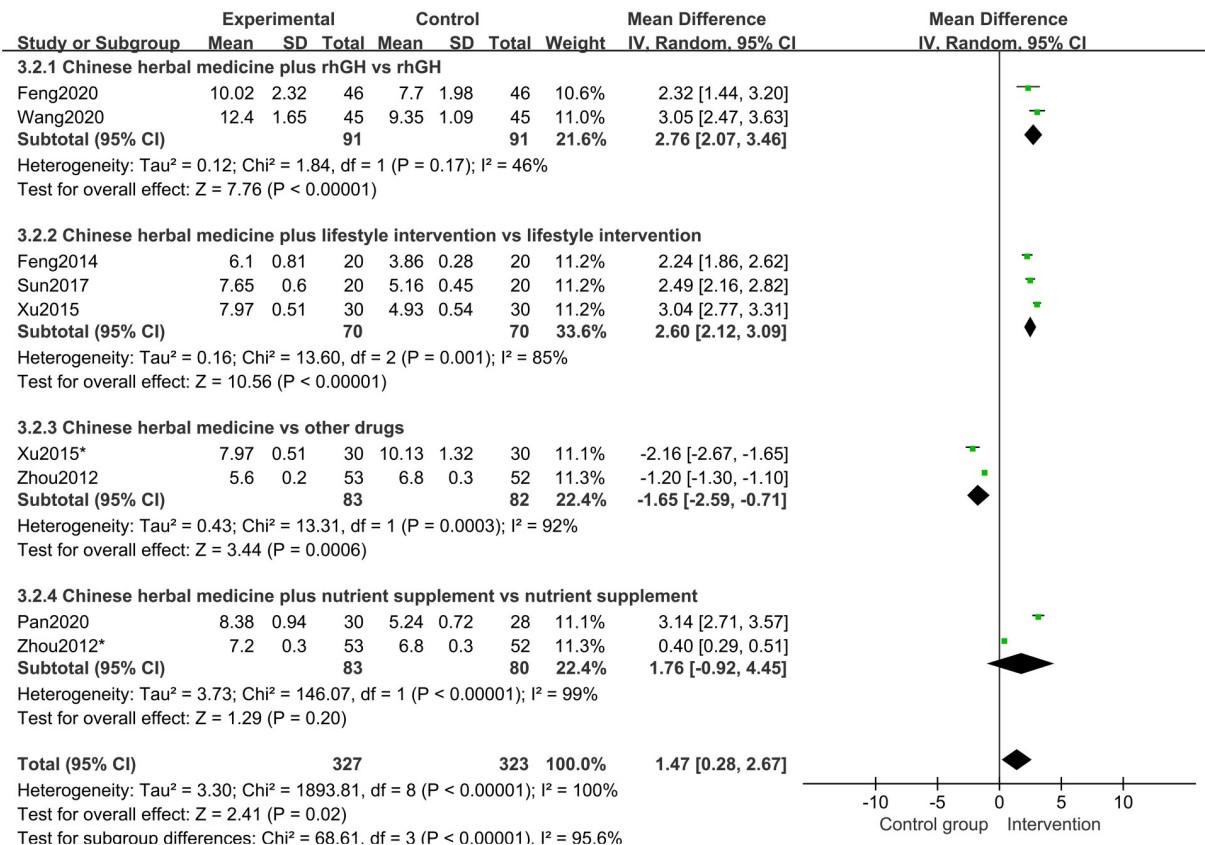

**Fig 8. Subgroup analysis of growth velocity based on the interventions.**

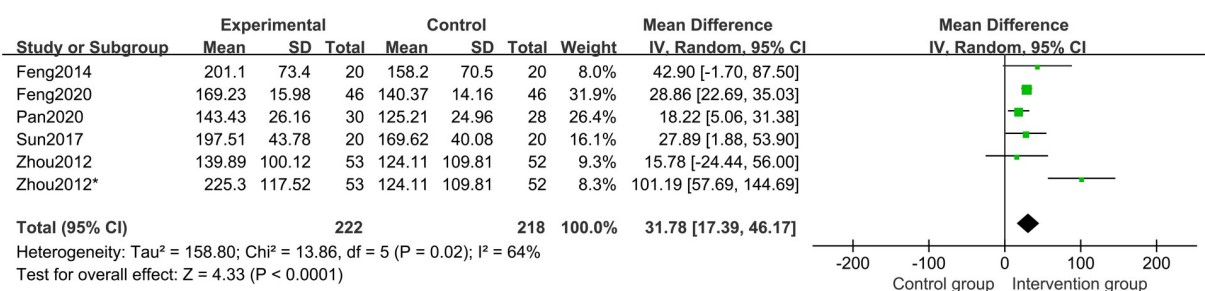

**Fig 9. Forest plot of IGF-1 level.**

comparisons indicated that CHM combined with other treatments could increase osteocalcin compared with other treatment alone groups (*P*<0.00001) (Fig 16).

Two studies [20, 22] of 80 participants reported GHP as an outcome. Fix effects model meta-analysis results showed that CHM plus lifestyle intervention could increase GHP level, compared with lifestyle intervention alone group (MD 3.29 points; 95%CI, 1.54 to 5.04; *P* = 0.0002; $I^2$ = 0%) (Fig 17).

**3.4.7 Differences in PAH between groups.** Two studies [20, 21] of 130 participants reported PAH as an outcome. Random-effects model meta-analysis results showed that there

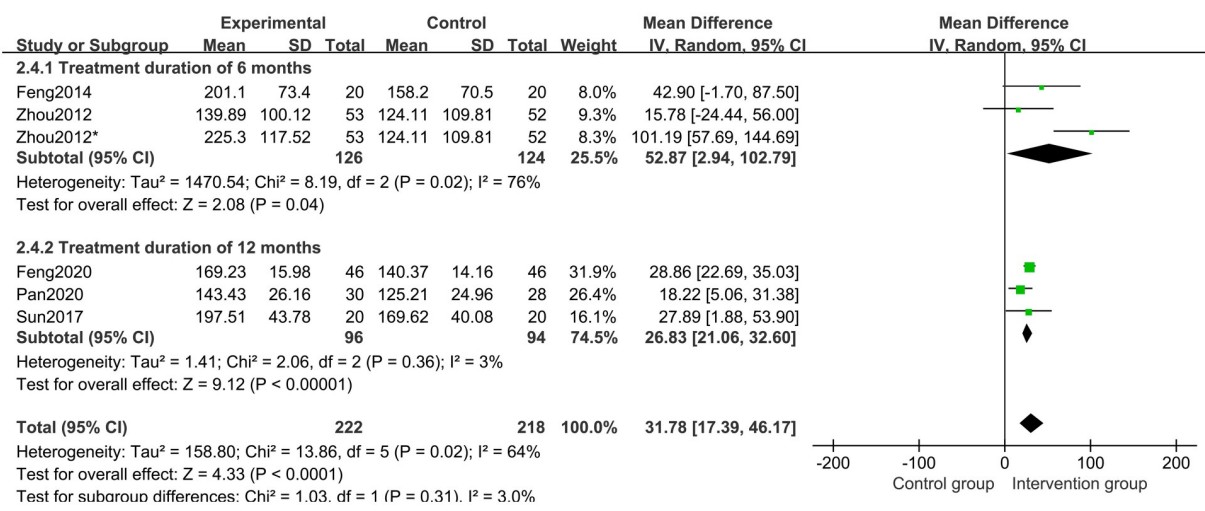

**Fig 10. Subgroup analysis of IGF-1 level based on the treatment duration.**

were no differences between groups in PAH (MD 3.14 points; 95%CI, -3.34 to 10.15; $P$ = 0.32; $I^2$ = 92%) (Fig 18). Subgroup analysis suggested that PAH in the CHM plus lifestyle intervention group was better than in the lifestyle intervention group alone ($P$<0.00001) (Fig 19).

**3.4.8 Side effects.** Out of the seven studies, only one [19] reported side effects (Transient headache, redness, skin swelling, diarrhea). Still, there were no differences between the intervention and control groups.

## 4. Discussion

There are no proven treatments for ISS, and GH treatment is expensive and controversial [8, 9]. This meta-analysis was the first attempt to evaluate the effectiveness of CHM with other ISS

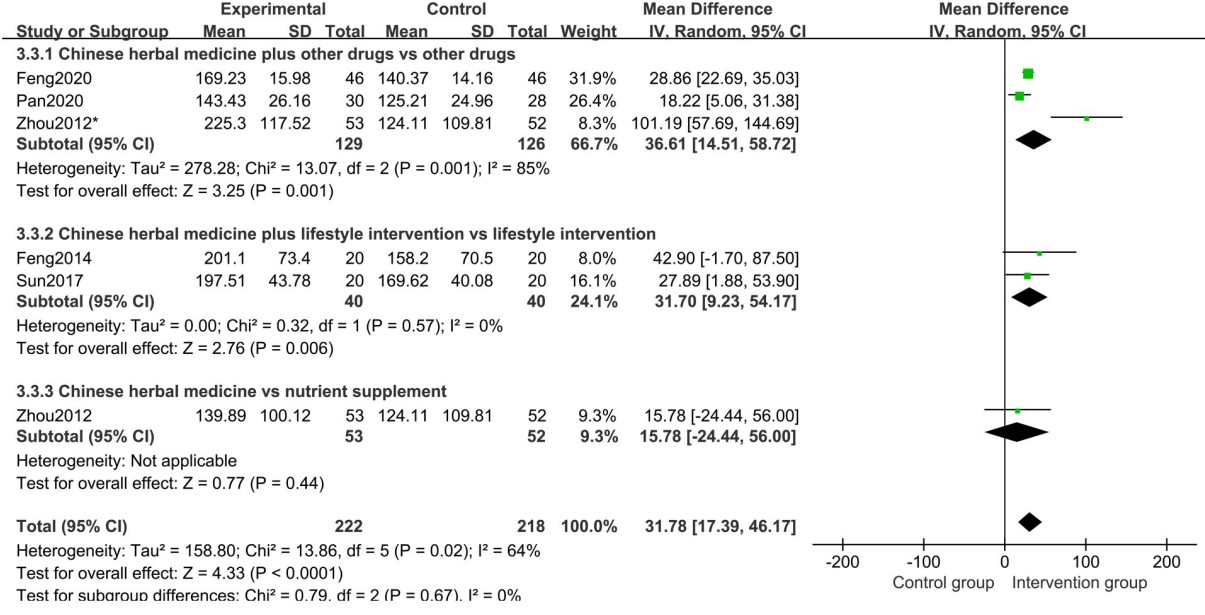

**Fig 11. Subgroup analysis of IGF-1 level based on the interventions.**

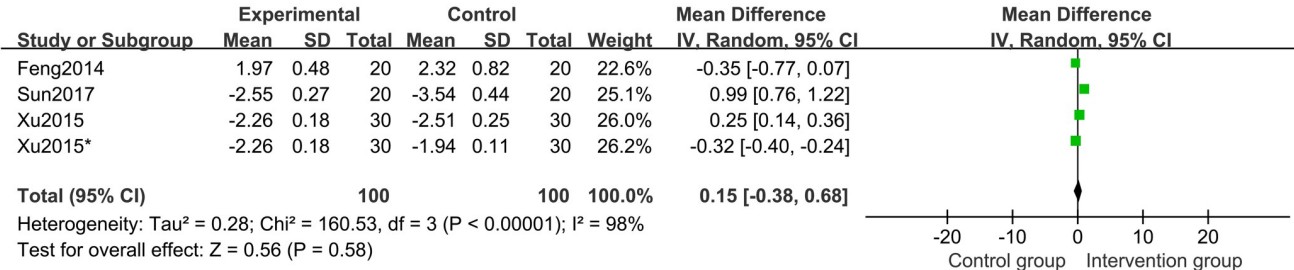

**Fig 12. Forest plot of height standard deviations score (HtSDS).**

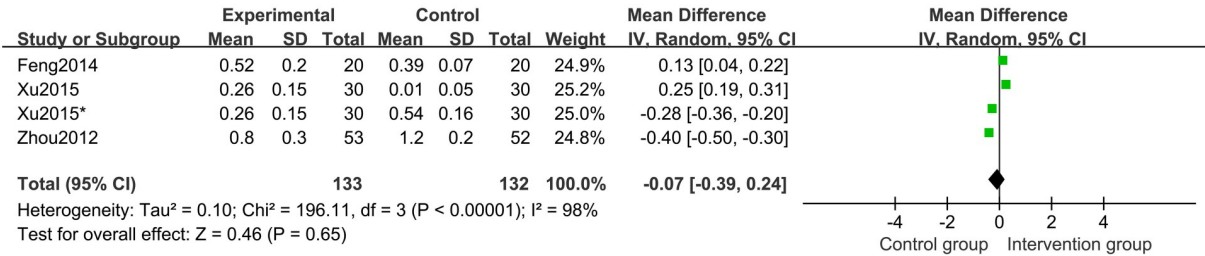

**Fig 13. Forest plot of change in HtSDS (ΔHtSDS).**

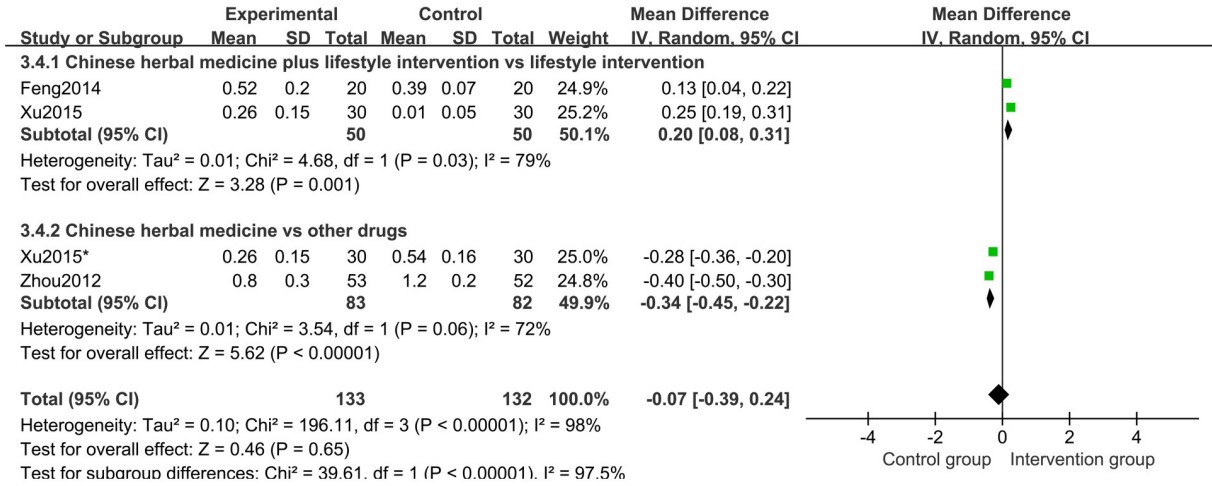

**Fig 14. Subgroup analysis of ΔHtSDS based on the interventions.**

treatments and CHM was discovered to have a similar effect in the clinic to other regularly utilized essential treatments which indicated that CHM is an optional treatment for clinical doctors and pediatricians.

## 4.1. Summary of main results

The presented meta-analysis included 7 trials to investigate the effects of CHM on Children with ISS. The major findings revealed that CHM was extremely effective at increasing height,

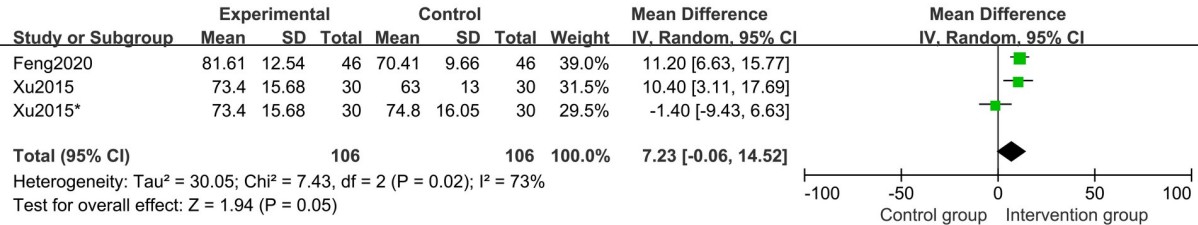

**Fig 15. Forest plot of osteocalcin.**

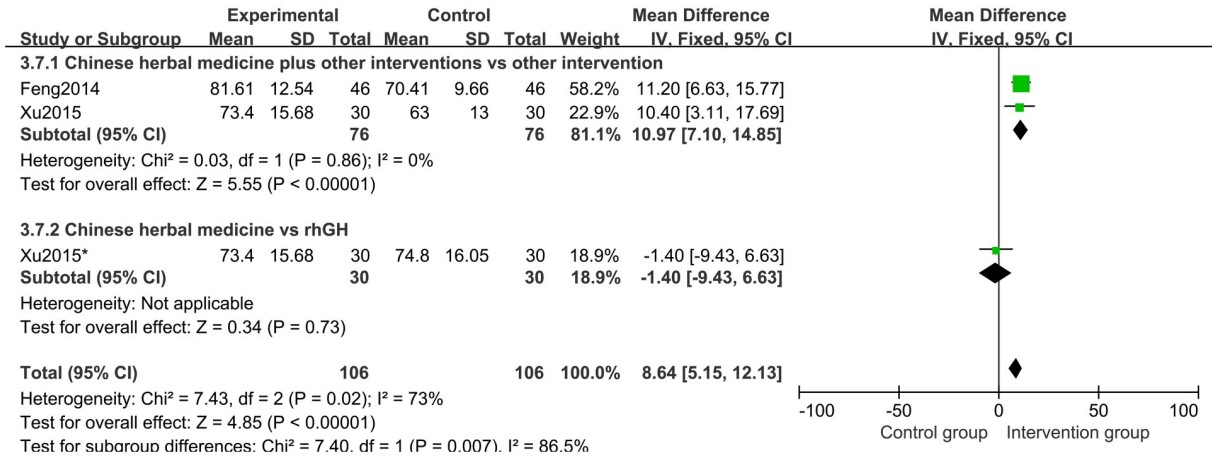

**Fig 16. Subgroup analysis of osteocalcin based on the interventions.**

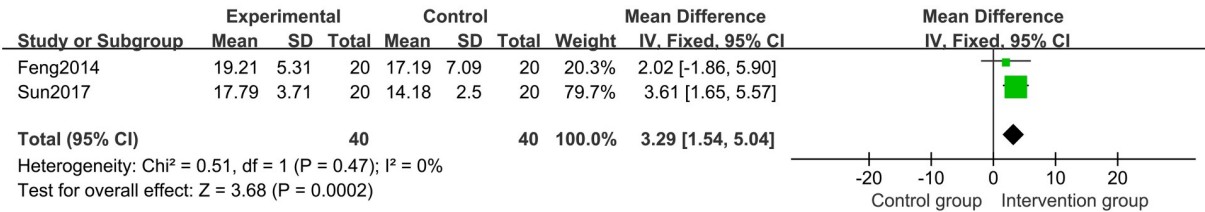

**Fig 17. Forest plot of the peak level of growth hormones (GHP).**

promoting growth velocity, raising IGF-1 level, and improving GHP. When compared to rhGH alone, CHM with rhGH had a substantial influence on enhancing growth velocity and IGF-1. When compared to lifestyle intervention alone, CHM plus lifestyle intervention had a substantial effect in increasing growth velocity, IGF-1, HtSDS, and PAH. It was discovered that combining rhGH with CHM was a more effective treatment for ISS than rhGH and life-style changes alone.

Many outcomes were all shown to be highly heterogeneous across control and experimental groups. However, finding the exact cause of heterogeneity was difficult because of the limited sample size of only seven studies included in the meta-analysis. As a result, we conducted sub-group analysis based on the types of comparisons, and duration of treatment, which revealed

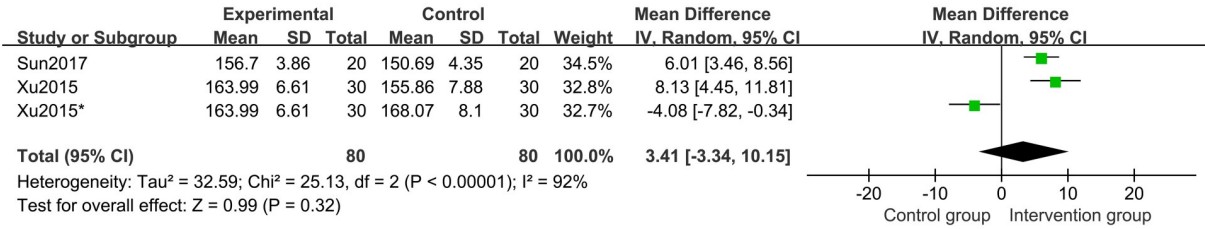

**Fig 18. Forest plot of predicting adult height (PAH).**

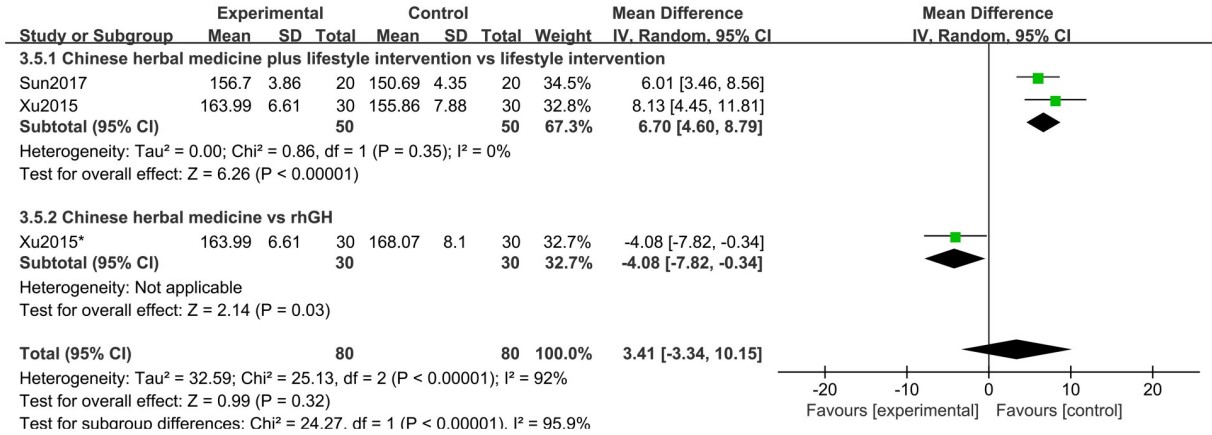

**Fig 19. Subgroup analysis of PAH based on the interventions.**

that the comparisons and treatment duration were not the primary drivers of high heterogeneity. Furthermore, the diverse CHM formulae employed in each trial may have an impact on the pooled results; nonetheless, six studies revealed six different CHM prescriptions utilized in the seven studies. As a result, we were unable to conduct a subgroup analysis based on CHM's prescriptions. We did not use a funnel plot to analyze the publication bias because the meta-analysis only contained seven papers.

In terms of the safety of CHM in ISS, none of the trials included in the study showed serious ISS side effects. Only one study in seven studies described some mild side effects, such as transient headache, red and swollen skin, and diarrhea, which were relived without additional treatment.

## 4.2 Limitations

However, this meta-analysis had some limitations. First, the sample size and the number of included research in this meta-analysis are modest, influencing the results' validity. Second, the included studies vary in terms of CHM dosage and formation, as well as the types of comparative treatment, treatment length, and so on. Furthermore, the data revealed that group heterogeneity was typically substantial, and despite conducting subgroup analysis, we were unable to find the reasons. Third, the studies included in this review are of poor quality, and the meta-analysis may have been influenced by the fact that all of the investigations were conducted in China. Overall, to get a more trustworthy conclusion, more high-quality studies must be conducted.

## 5. Conclusion

According to this systematic review and meta-analysis, CHM was found to be an effective treatment for ISS. On the one hand, CHM is as effective as other ISS treatments. On the other hand, compared with other treatments for ISS alone, CHM plus them can significantly benefit comprehensive clinical effect. For a reliable conclusion about CHM in the treatment of ISS, further evidence from large samples and high-quality RCTs is required to be investigated.

## Supporting information

**S1 Checklist. PRISMA 2009 checklist.**
(DOC)

## Author Contributions

**Conceptualization:** Yingying Li.

**Data curation:** Yingying Li, Xinying Chen, Zhengduo Liu.

**Formal analysis:** Yingying Li, Xinying Chen, Zhengduo Liu, Jinghua Yang.

**Funding acquisition:** Yingying Li, Jinghua Yang.

**Investigation:** Yingying Li.

**Methodology:** Yingying Li, Xinying Chen, Zhengduo Liu, Jinghua Yang.

**Project administration:** Yingying Li.

**Software:** Yingying Li, Xinying Chen, Zhengduo Liu.

**Supervision:** Xinying Chen, Jinghua Yang.

**Validation:** Xinying Chen, Jinghua Yang.

**Visualization:** Xinying Chen.

**Writing – original draft:** Yingying Li.

**Writing – review & editing:** Yingying Li, Xinying Chen.

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
