## [Decision Letter · Decision Letter 0]

11 May 2022

PONE-D-22-09803Chinese herbal medicine for children with idiopathic short stature (ISS): a systematic review and meta-analysisPLOS ONE

Dear Dr. Yang,

Thank you for submitting your manuscript to PLOS ONE. After careful consideration, we feel that it has merit but does not fully meet PLOS ONE’s publication criteria as it currently stands. Therefore, we invite you to submit a revised version of the manuscript that addresses the points raised during the review process.

We look forward to receiving your revised manuscript.

Kind regards,

Tariq Jamal Siddiqi

Academic Editor

PLOS ONE

Journal Requirements:

2. Please ensure that you include a title page within your main document. You should list all authors and all affiliations as per our author instructions and clearly indicate the corresponding author.

Reviewers' comments:

Reviewer's Responses to Questions

**Comments to the Author**

1. Is the manuscript technically sound, and do the data support the conclusions?

Reviewer #1: Yes

2. Has the statistical analysis been performed appropriately and rigorously? 

Reviewer #1: Yes

3. Have the authors made all data underlying the findings in their manuscript fully available?

Reviewer #1: Yes

4. Is the manuscript presented in an intelligible fashion and written in standard English?

Reviewer #1: Yes

5. Review Comments to the Author

Reviewer #1: 1. Please mention names of database in the abstract

2. Please mention company details with the software

3. Were the RCTs open-label, active or placebo-controlled? More details are required here

4. Major changes are required in the Methods section. The article needs to follow the PRISMA guidelines where the following sequence is suggested: Data Sources > Study Selection > Data Extraction > Data Synthesis

This change will improve the readability of the manuscript, as well as improving the quality of the paper.

5. Please mention mean age, Number of participants, Number of Males/Females and mean Study duration in the results section to improve the quality of the results.

6. The results are nicely written. However, please note that whenever the authors mention that some outcome has a better result than other, then this should be supported with a P-value. The p-values are not provided with any of the subgroup analysis results. Discussing significant and non-significant results in the Results section should also be supported with p-values.

7. The authors need to adequately discuss the strengths of the current findings. How this paper improves the medical literature and which gaps does the manuscript results cover. A brief discussion would improve the quality of the paper.

6. PLOS authors have the option to publish the peer review history of their article (what does this mean?). If published, this will include your full peer review and any attached files.

Reviewer #1: No

---

## [Author Response · Author response to Decision Letter 0]

1 Jun 2022

1st of June 2022

Dear reviewers:

Thank you for your specific and helpful comments. The manuscript has been improved according to the suggestions of the reviewer:

Major points

1. As suggested by the reviewer, we have added the names of the database in the abstract. 

2. The details of the software have been mentioned in the abstract and statistical analysis part.

3. We have added the details of the RCT types (study design section) in Table 1.

4. We have changed the sequence in the Methods section as per your suggestions.

5. The mean age, Number of participants, Number of Males/Females and mean Study duration were mentioned in Table 1. And we have also added a summary of these aspects of the included literature in the Research characteristics part.

6. We added the p-value in all the results that were able to be supported by a p-value.

7. The discussion of the strengths of the current findings of this paper is shown on lines 272 to 275.

Yours sincerely,

Jinghua Yang, MD

---

## [Decision Letter · Decision Letter 1]

12 Jun 2022

Chinese herbal medicine for children with idiopathic short stature (ISS): a systematic review and meta-analysis

PONE-D-22-09803R1

Dear Dr. Yang,

We’re pleased to inform you that your manuscript has been judged scientifically suitable for publication and will be formally accepted for publication once it meets all outstanding technical requirements.

Kind regards,

Tariq Jamal Siddiqi

Academic Editor

PLOS ONE

Additional Editor Comments (optional):

Reviewers' comments:

Reviewer's Responses to Questions

**Comments to the Author**

1. If the authors have adequately addressed your comments raised in a previous round of review and you feel that this manuscript is now acceptable for publication, you may indicate that here to bypass the “Comments to the Author” section, enter your conflict of interest statement in the “Confidential to Editor” section, and submit your "Accept" recommendation.

Reviewer #1: All comments have been addressed

2. Is the manuscript technically sound, and do the data support the conclusions?

Reviewer #1: Yes

3. Has the statistical analysis been performed appropriately and rigorously? 

Reviewer #1: Yes

4. Have the authors made all data underlying the findings in their manuscript fully available?

Reviewer #1: Yes

5. Is the manuscript presented in an intelligible fashion and written in standard English?

Reviewer #1: Yes

6. Review Comments to the Author

Reviewer #1: All of the concerns from the previous revision have been addressed by the authors. Interesting results about Chinese herbal treatment for children with idiopathic short stature are reported in this study. I congratulate the authors on their work being accepted for publication.

7. PLOS authors have the option to publish the peer review history of their article (what does this mean?). If published, this will include your full peer review and any attached files.

Reviewer #1: No

---

## [Editor Report · Acceptance letter]

16 Jun 2022

PONE-D-22-09803R1 

Chinese herbal medicine for children with idiopathic short stature (ISS): a systematic review and meta-analysis 

Dear Dr. Yang:

I'm pleased to inform you that your manuscript has been deemed suitable for publication in PLOS ONE. Congratulations! Your manuscript is now with our production department. 

Kind regards, 

on behalf of

Dr. Tariq Jamal Siddiqi 

Academic Editor

PLOS ONE